# Hyperacute Radiation Pneumonitis after Severe irAE

**DOI:** 10.3390/diagnostics14080850

**Published:** 2024-04-19

**Authors:** Yang Chou, Wei-Kai Chuang

**Affiliations:** 1Department of Otolaryngology-Head and Neck Surgery, Shuang Ho Hospital, Taipei Medical University, New Taipei 23561, Taiwan; 2Department of Otolaryngology-Head and Neck Surgery, Taipei Medical University Hospital, Taipei 11031, Taiwan; 3Department of Radiation Oncology, Shuang Ho Hospital, Taipei Medical University, New Taipei 23561, Taiwan; 4Department of Biomedical Imaging and Radiological Sciences, National Yang Ming Chiao Tung University, Taipei 11221, Taiwan

**Keywords:** immunotherapy, breast cancer, radiotherapy

## Abstract

A 54-year-old woman presented to an outpatient clinic with a recurrence of triple-negative breast cancer and multiple bone metastases. The patient had a large mass lesion of 10 cm on the sternum. She received the immune checkpoint inhibitors pembrolizumab and taxane. Initially, the patient responded excellently to treatment, but stopped pembrolizumab for grade IV skin toxicity with multiple ulcerative wounds over the bilateral leg and trunk. The lesions abated following administration of antibiotics and oral prednisolone for two months. After that, she was referred to the radiation oncology department for further treatment. She received radiotherapy for the sternum mass but stopped radiation at 42Gy/21 fractions for severe dyspnea and fever. Blood sampling found leukocytosis with neutrophil predominance. Chest radiography showed bilateral lung infiltration. Pulmonary CT scan yielded bilateral lung patchy consolidation compatible with radiation isodose-line. Bronchial lavage showed positive Pneumocystis jiroveci PCR. Dyspnea improved after titrating methylprednisolone within two days. The patient recovered well with TMP-SMX and glucocorticoids after the initiation of therapy.

The rapid development of pneumonitis following radiation therapy in this case, especially after the use of immunotherapy, is notably significant (Figure 1 and Figure 2). Typically, radiation pneumonitis occurs within a timeframe of 6 weeks to 6 months post-treatment, aligning with our reference [1,2]. However, in this case, following a Grade 3 immune-related adverse event from immunotherapy, the patient developed radiation pneumonitis at an unusually rapid pace after receiving radiation therapy, a clinical occurrence that is quite rare.

Immunotherapy, particularly the use of immune checkpoint inhibitors, has shown revolutionary effects in the treatment of various cancers. These drugs work by activating the patient’s immune system to attack cancer cells but can also lead to the immune system attacking normal tissues of the body, causing a range of immune-related side effects, including pneumonitis [3]. Recent retrospective studies have indicated that the combined use of ICIs and radiation therapy does indeed increase the incidence of related side effects [4,5,6]. In a pooled analysis examining the association of radiation therapy (RT) with the risk of adverse events in patients receiving immune checkpoint inhibitors (ICIs), Mitchell S. Anscher et al. found that administering an ICI within 90 days following RT did not significantly increase the risk of serious adverse events, suggesting the safety of this treatment sequence [6]. However, for patients who have experienced severe immune-related adverse events (irAEs), the risk of developing severe radiation pneumonitis remains unclear. 

The rapid progression of pneumonitis in this case could be attributed to the combined stimulation of the immune system by both the immune checkpoint inhibitor and radiation therapy. The serious consequences of potential opportunistic infections such as Pneumocystis jirovecii pneumonia resulting from the use of steroids should also be considered.

This occurrence also underscores the importance of close monitoring and evaluation of cancer patients undergoing immunotherapy to promptly identify and manage any potential pulmonary side effects. Furthermore, it highlights the need for personalized treatment plans for patients in cancer treatment, taking into consideration the balance between therapeutic benefits and potential side effects in treatment decisions.

## Figures and Tables

**Figure 1 diagnostics-14-00850-f001:**
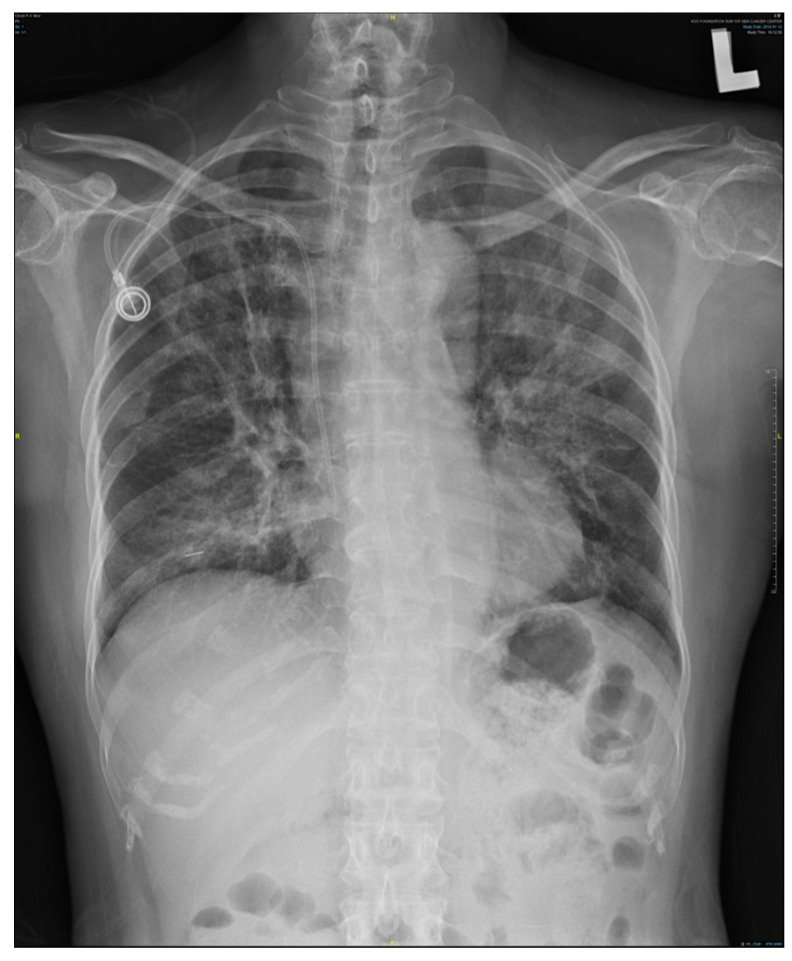
Chest X-ray showing marked pneumonitis patches, demonstrating increasing bilateral lung infiltration.

**Figure 2 diagnostics-14-00850-f002:**
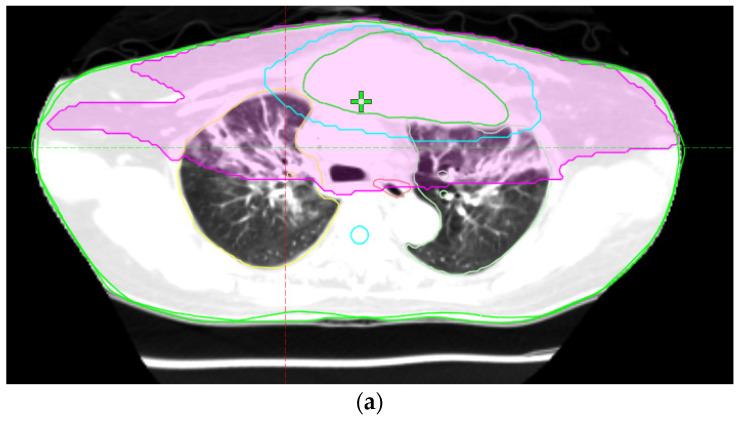
(**a**,**b**) Axial and sagittal views of a chest CT scan illustrating severe radiation pneumonitis. The images feature a fusion of radiotherapy isodose lines with the CT scan, highlighting areas where pneumonitis patches coincide with the isodose lines. The purple line represents the 20 Gy isodose line, and the sky-blue line represents the 42 Gy isodose line, indicating the direct impact of radiation dose distribution on lung tissue.

## Data Availability

The original data presented in the study are available on request from the corresponding author.

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
