# Peer review of "Hyperacute Radiation Pneumonitis after Severe irAE"

_diagnostics, 2024, doi:10.3390/diagnostics14080850_

Round 1
Reviewer 1 Report
Comments and Suggestions for Authors
The Authors may description concerning pneumocystis infection in the formation of abnormal opacities in upper lungs, because Immuno-check point -inhibitors may make latent PCP apparent. And had better describe whether the patient was administered cytotoxic drugs/glucocorticoid induce immuno-deficiency
Author Response
Thank you for your constructive feedback. In this case, the patient was administered steroids for two months following the use of Immune Checkpoint Inhibitors (ICIs). I have revised the original text to reorganize the sequence of information and concisely present this part.
Best regards,
Wei-Kai Chuang
Reviewer 2 Report
Comments and Suggestions for Authors
The reviewed short manuscript describes a clear clinical case of pneumonitis which has been developed as an adverse event after treatment of the advanced metastatic triple-negative breast cancer using immune checkpoint inhibitors (ICIs) and consequent radiation.
It seems that the description is too short. As the case report in the Non-published Material is written in Chinese I had no opportunity to get acquainted with it.
It should be noticed that there is a huge pool of publications on pneumonitis in patients treated only by ICIs, or only by lung irradiation, or by combined ICIs and radiation. Many publications are devoted to various methods including artificial intelligence for evaluation of the pneumonitis genesis.
It could be recommended to enhance the cited literature for better discussion of the observations obtained in this clinical case.
Comments on the Quality of English LanguageThe phrase “aligning with your reference” must be changed to “aligning with our reference”
Author Response
Thank you for your constructive feedback on our manuscript. We agree with your suggestion to expand the discussion of our case in the context of the extensive literature on pneumonitis related to immune checkpoint inhibitors (ICIs), radiation therapy, and their combination.
1. We will enhance our manuscript by including a broader review of relevant publications. We also do the editing for wrong English phrases
2. reply to "As the case report in the Non-published Material is written in Chinese I had no opportunity to get acquainted with it"
Non-published Material document is the Chinese version of the patient's informed consent form for the case report. Unfortunately, we cannot provide this document in English to reviewers due to the lack of an official English version. Please rest assured that the patient has given full informed consent for this research.
Best regards,
Wei-Kai Chuang
Round 2
Reviewer 1 Report
Comments and Suggestions for Authors
The revied manuscript is satisfactory
Reviewer 2 Report
Comments and Suggestions for Authors
The manuscript has been improved and is ready for publication